# Enantioselective reductive cross-couplings to forge C(sp²)–C(sp³) bonds by merging electrochemistry with nickel catalysis

Yun-Zhao Wang[1], Bing Sun[1], Jian-Feng Guo[1], Xiao-Yu Zhu[1], Yu-Cheng Gu [2], Ya-Ping Han [3], Cong Ma[1] & Tian-Sheng Mei [1] ✉

Motivated by the inherent benefits of synergistically combining electrochemical methodologies with nickel catalysis, we present here a Ni-catalyzed enantioselective electroreductive cross-coupling of benzyl chlorides with aryl halides, yielding chiral 1,1-diaryl compounds with good to excellent enantioselectivity. This catalytic reaction can not only be applied to aryl chlorides/bromides, which are challenging to access by other means, but also to benzyl chlorides containing silicon groups. Additionally, the absence of a sacrificial anode lays a foundation for scalability. The combination of cyclic voltammetry analysis with electrode potential studies suggests that Ni[I] species activate aryl halides via oxidative addition and alkyl chlorides via single electron transfer.

The 1,1-diaryl compounds represent a prevalent structural unit found in numerous pharmacologically active molecules and natural products (Fig. 1a)[1–3]. Consequently, the enantioselective synthesis of 1,1-diaryl compounds with high stereoselectivity holds meaningful significance. In recent years, Ni-catalyzed enantioselective cross-couplings have emerged as a robust strategy for constructing C(sp²)–C(sp³) bonds[4–9]. Compared to traditional asymmetric cross-coupling reactions, Ni-catalyzed reductive cross-couplings (RCCs)[10–15] offer an appealing approach for the enantioselective coupling of two electrophiles in the presence of a terminal reductant[16–20]. In this context, benzyl halides and styrene serve are crucial building blocks in the synthesis of 1,1-diarylalkanes[21–25]. For instance, Reisman et al. demonstrated Ni-catalyzed asymmetric RCCs of benzyl halides and with (hetero)aryl iodides using Mn as the reductant in 2017 (Fig. 1b, top)[21]. Furthermore, with the advent of photoredox/nickel dual catalysis, related asymmetric RCCs reactions have also been reported (Fig. 1b, bottom)[26,27]. However, the substrates for the aforementioned reactions are limited to aryl iodides or highly reactive aryl bromides. The reductive cross-coupling of aryl chlorides or electron-rich aryl bromides with benzyl chlorides remains a significant challenge, presumably due to the lower reactivity of aryl chlorides or electron-rich aryl bromides with the nickel catalyst compared with aryl iodides[28,29].

With the renaissance of organic electrolysis, it showcases potent advantages in tunability, sustainability, and scalability[30–42]. In this regard, enantioselective nickel-catalyzed electroreductive cross-couplings (eRCCs) have emerged as a promising strategy for achieving asymmetric transformations, involving the regeneration of low-valent nickel species through cathodic reduction as exemplified by Reisman[43], Mei[44–46], and Nevado[47]. In 2023, our group reported a Ni-catalyzed enantioselective eRCCs of acrylates with aryl chlorides/bromides and alkyl bromides, which provided a feasible means to activate aryl chlorides or electron-rich aryl bromides with low-valent nickel species[48].

In this study, we present the Ni-catalyzed enantioselective eRCCs of benzyl chlorides with aryl halides, affording chiral 1,1-diaryl compounds with good to excellent enantioselectivity (Fig. 1c). Specifically, this reaction exhibits a broad substrate scope and is applicable for silicon-substituted benzyl chlorides or aryl chlorides which have not been previously reported in the literature. Furthermore, this electroreductive strategy is performed without the need for heterogeneous metal reductants or sacrificial anodes. Instead, it utilizes constant current electrolysis in an undivided cell, with triethylamine acting as the terminal reductant[49,50].

[1]State Key Laboratory of Organometallic Chemistry, Shanghai of Organic Chemistry, University of Chinese Academy of Sciences, Chinese Academy of Sciences, Shanghai, PR China. [2]Syngenta, Jealott's Hill International Research Centre, Berkshire, UK. [3]School of Chemical Engineering and Technology, Hebei University of Technology, Tianjin, China. ✉e-mail: mei7900@sioc.ac.cn

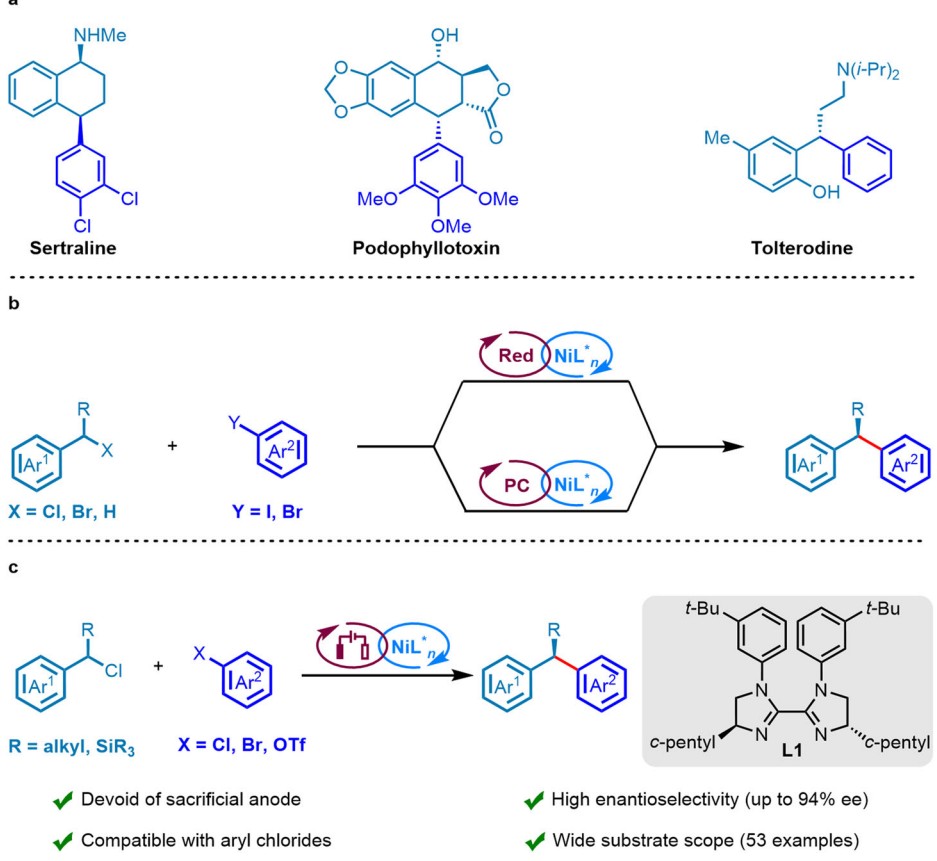

**Fig. 1 | Ni-catalyzed enantioselective cross-couplings for the synthesis of 1,1-diaryl compounds. a** Examples of bioactive 1,1-diaryl drugs. **b** Nickel-catalyzed enantioselective cross-couplings (prior strategies). **c** This work: Enantioselective electroreductive cross-couplings.

## Results

### Optimization studies

Initially, we investigated the asymmetric arylation of ethylbenzyl chloride (**1a**) with methyl 4-bromobenzoate (**2a**) using different chiral ligands (Table 1). In recent years, chiral biimidazoline (BiIM) ligands have been widely used in Ni-catalyzed asymmetric RCCs reactions[51–55]. The application of BiIM ligands in reactions generally resulted in good yields and enantioselective conversions (**L1**–**L6**). For instance, employing **L1** as the ligand resulted in the formation of chiral 1,1-diaryl product **3a** in 89% yield and 90% ee (entry 1). Contrastingly, the use of bisoxazolines (Biox) as ligands led to lower ee values for product **3a** (entries 7 and 8). This can be attributed to the greater electron richness exhibited by BiIM ligands compared to Biox ligands, thereby enhancing catalyst reactivity under electrochemical conditions[46]. Besides, the addition of 4 Å molecular sieves as desiccants significantly enhances the reaction. In addition, when DMAc was used as the solvent instead of mixed solvent (DMAc:THF = 1:45), a decrease in ee value was observed (entry 9). This underscore the significance of using mixed solvents to mitigate solvation effects, thereby enhancing reaction enantioselectivity[56,57]. We employed RVC or C as the anode instead of Pt, and RVC as the cathode instead of Ni form, resulting in moderate yields (entries 10–12). Control experiments confirmed the indispensability of nickel, **L1**, and electric current for this asymmetric transformation (entry 13).

With the optimized reaction conditions established, we proceeded to explore the generality and limitations of this Ni-catalyzed enantioselective *e*RCCs of benzyl chlorides with aryl halides. As detailed in Fig. 2, various aryl halides efficiently underwent the desired enantioselective arylation reactions. *Para*-substituted aryl chlorides bearing diverse functional groups such as ester (**3a**, **3b**, **3 f**), sulfonyl (**3c**), cyano (**3 d**), aryl (**3e**) groups were well tolerated under

electrochemical conditions, yielding the corresponding aryl ester products in good yields and high enantioselectivities. Additionally, aryl triflates could be utilized as electrophiles, affording products **3a** and **3e** with yields of 63% and 45%, and ee values of 82% and 92%, respectively. However, this reaction was limited to chlorobenzenes attaching electron-deficient substituents, showing low activity with substrates containing electrically neutral or electron-rich substituents[48].

In addition to chloride electrophiles, we investigated aryl bromides as coupling partners. Bromobenzene with electron-deficient substituents afforded the products in good yields and enantioselectivities (**3a**–**3e**, **3 g**, **3l**–**3n**, **3w**–**3x**), with the absolute configuration of the product **3 g** determined by single-crystal X-ray diffraction. Moreover, the reaction was compatible with aryl bromides bearing electron-rich/neutral substituents in the para- or meta-position, yielding products with good yields and moderate to good ee values (**3h**–**3k**). Subsequently, we explored polyaromatic bromines including naphthalene bromides (**3o** and **3p**) and bromofluorene (**3q**), obtaining good yields and ee values. Furthermore, various heterocyclic bromobenzenes served as coupling partners, yielding products with moderate to good yields and excellent ee values (**3r**–**3 v**). However, pyridine and pyrimidine bromobenzenes proved to be less viable substrates, producing the corresponding products in moderate yields and ee values (**3 y** and **3z**).

Benzyl chloride bearing various functional groups on the aromatic ring were also found to be a suitable substrate for these *e*RCCs, affording products **3aa**–**3al** with good yields and ee values. In addition, substituting the benzylic ethyl group with other alkyl groups maintained the reaction's high yield and enantioselectivity (**3 f**, **3af**–**3aj**). Interestingly, replacing the alkyl group with a silicon substituent[24] also resulted in successful conversion, yielding products **3ak** and **3al** with good yields and ee values. To further investigate the synthetic

## Table 1 | Reaction optimization[a]

| Entry | Variations from standard conditions | Yield (%)[b] | ee (%) |
|---|---|---|---|
| 1 | none | 89(85) | 90 |
| 2 | L2 instead of L1 | 83 | 85 |
| 3 | L3 instead of L1 | 78 | 72 |
| 4 | L4 instead of L1 | 84 | 87 |
| 5 | L5 instead of L1 | 75 | 72 |
| 6 | L6 instead of L1 | 83 | 68 |
| 7 | L7 instead of L1 | 67 | 18 |
| 8 | L8 instead of L1 | 34 | 25 |
| 9 | DMAc instead of DMAc:THF = 1:45 | 92 | 80 |
| 10 | Anode RVC instead of Pt | 49 | 89 |
| 11 | Anode C instead of Pt | 57 | 85 |
| 12 | Cathode RVC instead of Ni foam | 35 | 87 |
| 13 | w/o [Ni], L1, or electric current | — | — |

[a]Reactions were carried out with benzyl chloride 1a (2 equiv.), aryl bromide 2a (0.2 mmol), NiBr$_2$•glyme (10 mol%), L1 (15 mol%), TBABF$_4$ (1 equiv.), Et$_3$N (4.5 equiv.), 4 Å MS (50 mg), DMAc:THF = 1:45 (2 ml), and platinum (1 × 1 cm$^2$) as the anode. The Ni form (1 × 2.5 cm$^2$) was as the cathode, 2 mA, room temperature, 12 h. [b]Yields determined by $^1$H NMR using CH$_2$Br$_2$ as the internal standard. Isolated yield is shown in parentheses.

application of Ni-catalyzed enantioselective eRCCs between benzyl chlorides and aryl halides, we synthesized analogs of naproxen (3am), adamantanemethanol (3an), cholesterol (3aq), menthol (3ar), and diacetone-D-galactose (3as) with good yields and excellent enantioselectivities. Notably, we efficiently synthesized an anti-smallpox agent (3ao) and a sertraline intermediate (3ap), thus demonstrating the effectiveness of this enantioselective methodology in producing a range of biologically active compounds.

Interestingly, this electrochemical reductive coupling demonstrates complementary reactivity with traditional reductive coupling methods. When manganese (Mn) as a reductant in the traditional approach, it shows promise for reactions involving pyridine iodides but struggles with the efficient conversion of pyridine bromides and other bromobenzene (Fig. 3A, B). On the other hand, electrochemistry emerges as a more adaptable method for bromobenzene, capable of effectively incorporating various substituents. Despite this, pyridine halides present a challenge for electrochemistry as well, yielding lower rates of conversion and ee values, as indicated in Fig. 3A, B. This inefficiency is mainly due to the destructive effect of the pyridine ring on the binding of nickel to the ligand during the reaction. What stands out, however, are the findings from time-course experiments that showcase electrochemical conditions favoring both the production of compound 3a and the consumption of substrate 2a over the reactions

employing Mn as the reductant, as demonstrated in Fig. 3C. This suggests that, despite the limitations faced by each method when dealing with certain substrates, electrochemistry holds a distinct advantage in terms of reaction kinetics, offering a pathway to more efficient syntheses under selected conditions.

The oxidative addition of aryl halides to low-valent nickel species is a critical step in eRCCs. The collective research, including that from previous studies and our group, supports the involvement of Ni$^I$ species in reacting with aryl halides[58–61]. Through cyclic voltammetry (CV) analysis, it's observed that the mixture NiBr$_2$•glyme and L1 at a 1:1 ratio shows two quasi-reversible reductive peaks at −1.68 V and −2.19 V vs. Fc$^+$/Fc$^0$ in the solution (DMAc:THF = 1:45). These peaks are attributed to the reductive potential of Ni$^{II}$/Ni$^I$ and Ni$^I$/Ni$^0$, respectively (Fig. 3D)[62,63]. Notably, the presence of substrate 2a results in a significant increase in the reduction peaks of Ni$^{II}$/Ni$^I$, accompanied by the disappearance of the oxidation peak at -0.55 V, suggesting the oxidative addition of 2a to the Ni$^I$ species. Further exploration using 1a for CV analysis in the NiBr$_2$•glyme and L1 mixture indicated a substantial increase in the Ni$^{II}$/Ni$^I$ reduction peak, hinting at the generation of benzyl radicals through a single electron transfer (SET) from Ni$^I$ species (Fig. 3E)[64,65]. To exclude alternative pathways for benzyl radical formation, such as halogen atom transfer (XAT) with benzyl chloride and anodized

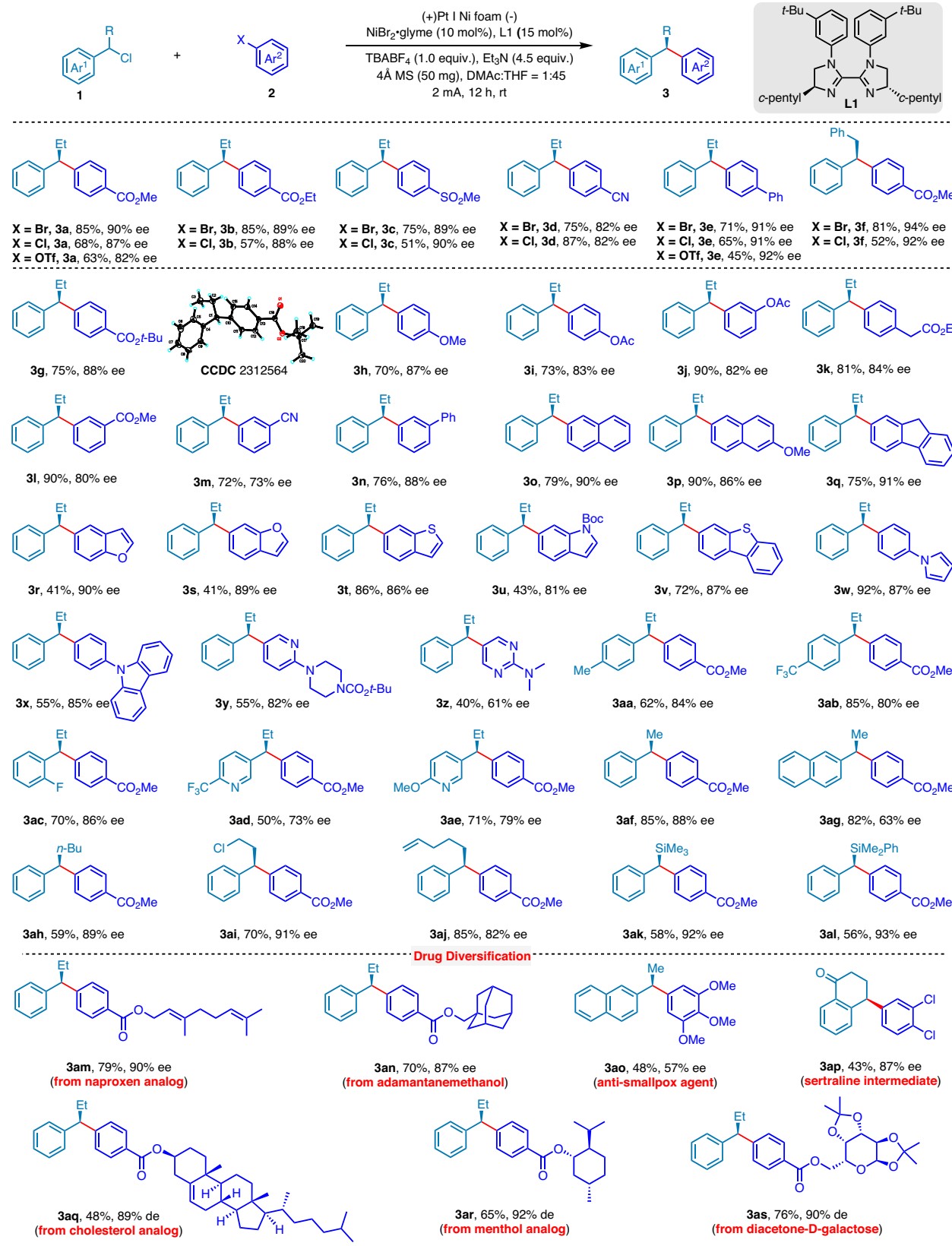

**Fig. 2 | Substrate scope.** All yields refer to isolated products.

triethylamines[66–68], additional CV analysis with **1a** and Et₃N was conducted, revealing no significant change in the oxidation peak of Et₃N (+0.63 V), thus supporting the proposed mechanism (Fig. S3 in Supplementary information).

The investigation further delves into the mechanism of these Ni-catalyzed *e*RCCs by employing radical clocks like **1b** (Fig. 3F). The use of a cyclopropyl-containing compound **1b** resulted in a ring-opened product **4a** with 48% yield, reinforcing the theory that the reaction

pathway likely involves radicals. Additionally, the absence of a sacrificial anode highlights the potential for scaling up this reaction, with the gram-scale synthesis of **3a** achieving an 83% yield and 90% enantioselectivity (Fig. 3G).

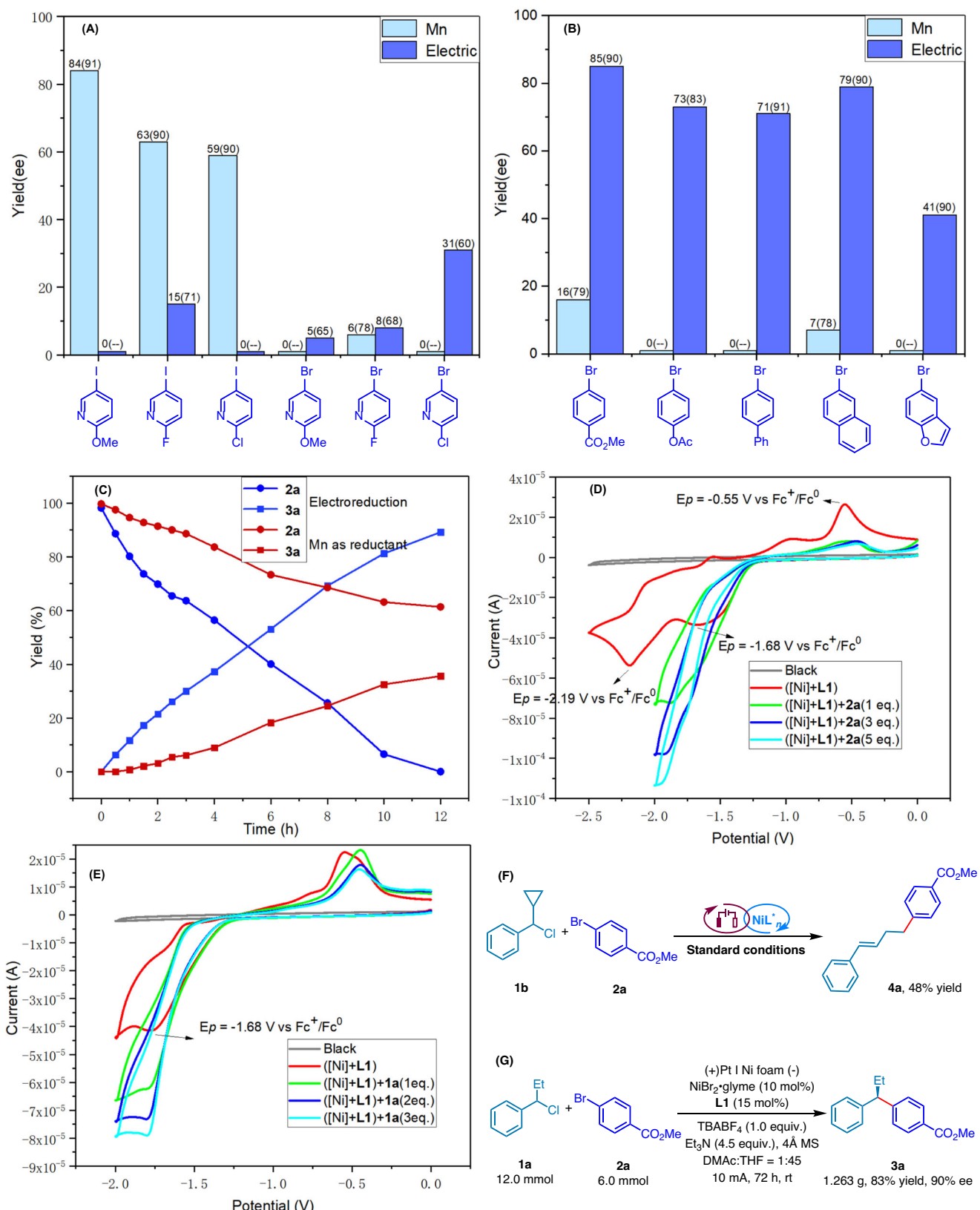

**Fig. 3 | Mechanistic experiments. A** Comparison of reactions of pyridine halides under different reducing conditions. **B** Comparison of reactions of aryl bromides under different reducing conditions. **C** Time course experiment. **D** CV analysis on the interaction of **2a** with the catalyst. **E** CV analysis on the interaction of **1a** with the catalyst. **F** Radical clock. **G** Gram-scale preparation.

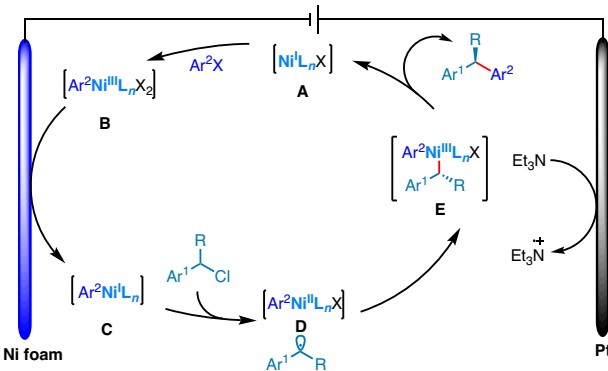

**Fig. 4 | Proposed mechanism.** Proposed Ni-catalyzed enantioselective *e*RCCs mechanism.

Based on these studies and previous reports[69–71], a proposed mechanism is presented for the Ni-catalyzed enantioselective *e*RCCs, beginning with the formation of [Ni[I]] species **A** via cathodic reduction of a [Ni[II]] precatalyst (Fig. 4). The oxidative addition of aryl halides to this species yields an Ar[Ni[III]] species **B**, which then undergoes cathodic reduction to generate Ar[Ni[I]] species **C**. This species can activate benzyl chloride, generating benzyl radical intermediate and Ar[Ni[II]] species **D**. The radical intermediate is subsequently trapped by **D** to form [Ni[III]] intermediate **E**, and reductive elimination from **E** yields the cross-coupled product and regenerates [Ni[I]] **A**, completing the catalytic cycle. Notably, the oxidation of triethylamine primarily occurs at the anode, facilitating the overall reaction process.

In summary, we have presented the Ni-catalyzed enantioselective *e*RCCs of benzyl chlorides with aryl halides in an undivided cell, affording a wide range of chiral 1,1-diaryl compounds under mild reaction conditions. This innovative enantioselective *e*RCC approach applies to aryl chlorides/bromides, which is difficult to achieve by other means. Additionally, we have demonstrated the compatibility of this reaction with silicon-substituted benzyl chloride substrates. Further efforts to develop catalytic enantioselective *e*RCCs of unactivated alkyl halides are currently underway in our laboratory.

## Methods
### General Procedure for electrochemical reaction
In glovebox, an oven-dried electrochemical cell with a stir bar was charged with bromobenzene/chlorobenzene/aryl triflates (**2**, 0.2 mmol, 1 equiv.) and benzyl chloride (**1**, 0.4 mmol, 2 equiv.), NiBr$_2$·glyme (0.02 mmol, 10 mol%), ligand **L1** (0.03 mmol, 15 mol%), TBABF$_4$ (0.2 mmol, 1 equiv.), Et$_3$N (0.9 mmol, 4.5 equiv.), 4 Å MS (50 mg), 2 mL of DMAc:THF = 1:45. The tube was installed an Ni foam as the cathode and Pt as the anode. The mixture was stirred at room temperature for 30 min. The reaction mixture was electrolyzed under a 2 mA at RT. After 12 h, EtOAc (50 mL) was added to the resulting solution, which was then washed with brine (50 mL × 3). The organic layer was dried over anhydrous Mg$_2$SO$_4$, filtered and concentrated to give the crude product. The crude product was purified by automated silica gel column chromatography (EtOAc/hexanes) to afford the desired product **3**. More experimental procedures and photographic guide for enantioselective reductive cross-couplings to forge C(sp$^2$)–C(sp$^3$) bonds by merging electrochemistry with nickel catalysis are provided in Supplementary Information.

## Data availability
The X-ray crystallographic coordinates for structures reported in this article have been deposited at the Cambridge Crystallographic Data Center (CCDC), under deposition number CCDC 2312564 (3 g). The data can be obtained free of charge from the Cambridge Crystallographic Data Center [http://www.ccdc.cam.ac.uk/data_request/cif].

The data generated in this study are provided in the Supplementary Information files. Data supporting the findings of this manuscript are also available from the corresponding author upon request.

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

## Acknowledgements

This work was financially supported by National Key R&D Program of China (No. 2021YFA1500100), the Strategic Priority Research Program of the Chinese Academy of Sciences (XDB0610000), the NSF of China (Nos. 21821002, 22361142834, 22425111, and 22101294), the S&TCSM of Shanghai (21ZR1476500), and Natural Science Foundation of Ningbo (2023J035).

## Author contributions

T.M. designed and directed the project. Y.W. performed the experiments and developed the reactions. B.S., J.G., and X.Z. helped collecting some experimental data and prepared the supplemental information. Y.W., Y.G., Y.H., and C.M. prepared the manuscript. All authors discussed the results and commented on the manuscript.

## Competing interests

The authors declare no competing interests.
