## [Transparent Peer Review file · Nature Communications]

Enantioselective Reductive Cross-Couplings to Forge C(sp²)–C(sp³) Bonds by Merging Electrochemistry with Nickel Catalysis

Corresponding Author: Professor Tian-Sheng Mei

Version 0:

Reviewer comments:

Reviewer #1

(Remarks to the Author)

In this manuscript, Mei and co-workers report the electrochemical promoted nickel catalysis for the enantioselective reductive cross-coupling of benzyl halides with aryl halides.

As mentioned by the authors in this manuscript, such type of reactions has also been disclosed before by using thermal or photo conditions, albeit with relatively narrow substrate scope limitation. Herein, the authors tried to challenge such reaction by using electrochemical promoted nickel catalysis, and the results is attractive.

Please solve the following questions.

- (1) Since Et₃N has been used as the terminal reductant, please specific it in Scheme 1, and replace it with the claim “electric current as the reductant”
- (2) Please specify Scheme 1b, bottom in the main context.
- (3) Why molecular sieve is necessary in this system? please specify it in the main content.
- (4) Since aryl triflate gave lower results, how about the use of ArOTs, as this compound is more stable under electrochemical conditions and can do the oxidative addition with nickel easily.
- (5) In line 137, please cite the evidence shown in SI.
- (6) As for the reaction mechanism, how to rule out the possibility that benzylic radical was formed via the direct reduction of LNi(I), as Figure 1E has shown the strong evidence? This is the biggest problem for this manuscript.
- (7) Since the authors have claimed the synthetic importance of products, it is possible to give some examples?
- (8) Since the chiral ligand is expensive, is it possible to decrease the amount or recovery it at gram scale?
- (9) Since thermal conditions using Mn as the reductant has shown the advantage in some cases, it seems unfair to claim it in line 105. Please revise it properly. In addition, please specify the source for Mn, and nickel source.
- (10) As for supplementary table S8. screening of additive, please show the detail. As for the SI, please specify how to reuse the electrodes.

Reviewer #2

(Remarks to the Author)

Mei and colleagues described a novel methodology for the asymmetric synthesis of optically active 1,1-diaryls through enantioselective electrochemical nickel catalyzed cross coupling of benzylic chlorides with aryl chlorides, triflates and bromides. This innovative enantioselective eRCC approach applies to aryl chlorides/bromides, which is difficult to achieve by other means. Chiral 1,1-diaryls generation protocols continue to attract significant attention from the community due to their pharmacologically active molecules and natural products. The compatibility of the enantioselective reductive cross-couplings protocol was investigated under mild reaction conditions, including a scope of 45 entries containing the asymmetric 1,1-diaryl motif and 7 examples that represent drugs or analogs thereof, which turned out to be fairly general and avoided the employment of heterogeneous metal reductants or sacrificial anodes. Furthermore, comparison studies with previously reported conditions employing Mn as a reductant was carried out to highlight their methods compatibility, and a mechanism of this protocol can be constructed from the results of well-organized experimental mechanistic studies and cyclic voltammetry experiments.

Overall, in my impression the work presented in this manuscript has high conceptual innovation and uniqueness that

warrants publication on Nature Communications. The manuscript is well-organized, with experimental procedures well-described and compounds adequately characterized. I am thus enthusiastic to recommend publication of the manuscript after some minor issues are addressed.

1) The authors cited e.g., Zhan Lu's enantioselective benzylic arylation via photoredox and nickel dual catalysis, this work utilized an almost similar ligand to the one reported in Lu's work. What is the difference between the two works?

2) In table 2. substrate scope, are other electron-poor (hetero)aryl chlorides such as chloro-pyrimidines, pyridazines, pyrazines, or triazines compatible with the optimal reaction conditions?

3) Plenty of work underwent in terms of ligand design for enantioselective couplings since Reisman's report in 2017 and Lu's in 2019. How does Reisman's procedure from 2017, which relies on a BiOX ligand that was notable for its time, work with the new BiIM system?

4) Does the use of platinum and the subsequent use of TEA have a negative effect on the catalysis, through catalyst deactivation via coordination of TEA to nickel?

Version 1:

Reviewer comments:

Reviewer #1

(Remarks to the Author)

This reviewer has carefully checked the revised manuscript and is fully satisfied with the revisions. Therefore, I am very happy to recommend it for publication in this journal at the current form.

Reviewer #2

(Remarks to the Author)

All the questions has been carefully addressed, this work can be published as current form.

Response to Reviewers and Editors

(Manuscript No.: NCOMMS-24-60444-T)

I. Response to Reviewers

II. Response to Editors

I. Response to Reviewers

Reviewer: 1

Question 1: Since Et₃N has been used as the terminal reductant, please specify it in Scheme 1, and replace it with the claim “electric current as the reductant”

Our response: Thanks for this great suggestion. “Et₃N as the terminal reductant” implies that Et₃N can serve as a sacrificial agent at the anode to prevent the degradation of the anode metal. “Electric current as the reductant” indicates that nickel can be reduced on the electrode surface. Since these two concepts convey different meanings, we have removed “electric current as the reductant” from Scheme 1.

Question 2: Please specify Scheme 1b, bottom in the main context.

Our response: Thanks for this suggestion. We have deleted the Scheme 1b bottom.

Question 3: Why molecular sieve is necessary in this system? please specify it in the main content.

Our response: After adding 4 Å molecular sieves to the reaction, the yield can be increased from 39% to 89% (see **Table S7, S8** in Supplementary Information). Accordingly, the addition of 4 Å molecular sieves as desiccants significantly enhances the reaction.

Supplementary Table S7. Increase the substrate resistance reaction

Supplementary Table S8. Screening of additive

Question 4: Since aryl triflate gave lower results, how about the use of ArOTs, as this compound is more stable under electrochemical conditions and can do the oxidative addition with nickel easily.

Our response: Thanks for this valuable suggestion. We synthesized the following ArOTs substrates and incorporated them into the reaction; however, we observed no product formation. The ArOTs remained largely unchanged throughout the reaction, suggesting that it did not undergo oxidative addition with the low-valent nickel as shown in the following table.

Entry	R	NMR yield (%)	2 (%) ^c
1	CO ₂ Me	0	-
2	NO ₂	0	-
3	CN	0	-
4	Ph	0	-

Question 5: In line 137, please cite the evidence shown in SI.

Our response: Thanks for this valuable comment. We have added "(Fig. S3 in Supplementary information)" to the corresponding position.

Question 6: As for the reaction mechanism, how to rule out the possibility that benzylic radical was formed via the direct reduction of LNi(I), as Figure 1E has shown the strong evidence?

Our response: Thanks for this comment. Figure 1D and 1E illustrate that $L_nNi(I)$ species can undergo both oxidative addition with aryl halides and single electron transfer with benzyl chloride to generate radicals. However, this alone does not fully account for the efficient cross-coupling observed in the reaction. In conjunction with the relevant mechanistic studies by Diao, Doyle, and Sigman (*J. Am. Chem. Soc.* **2023**, *145*, 20551; *J. Am. Chem. Soc.* **2022**, *144*, 5575; *J. Am. Chem. Soc.* **2023**, *145*, 8689), we posit that this mechanism provides a more comprehensive explanation for why this reaction achieves highly efficient cross-coupling. Specifically, the oxidative addition of $L_nNi(I)X$ species to aryl halides, followed by reaction of the resulting $L_nNi(I)Ar$ species with benzyl chloride to generate radicals, likely plays a crucial role.

Question 7: Since the authors have claimed the synthetic importance of products, it is possible to give some examples?

Our response: Sertraline is a drug molecule to treat depression and obsessive-compulsive disorder. Traditional synthesis methods involved the resolution of diastereomeric reduced amines, which were challenging to separate due to their two chiral centers, resulting in low yields. In contrast, this electrochemical reaction can directly produce the intermediate **3ap** with high enantioselectivity as illustrated in the following scheme.

Question 8: Since the chiral ligand is expensive, is it possible to decrease the amount or recovery it at gram scale?

Our response: Thanks for this suggestion. We conducted the following reaction as detailed in the following table. When the ligand quantity was reduced to 10 mol%, the yield of the reaction decreased significantly (entry 2). Additionally, when both the nickel and ligand equivalents were reduced simultaneously, the yield still decreased substantially (entries 3 and 4). This indicates that the catalytic efficiency of the reaction is low. The ligand remains intact during the reaction and can be recycled on a gram scale for reuse. Therefore, if the catalytic efficiency can be enhanced, the reaction cost can be significantly reduced. This challenge is likely shared by most Ni-catalyzed reductive cross-couplings reactions.

Entry	Variations from standard conditions	NMR yield (%)
1	NiBr ₂ ·glyme (10 mol %), L1 (15 mol %)	89
2	NiBr ₂ ·glyme (10 mol %), L1 (10 mol %)	35
3	NiBr ₂ ·glyme (5 mol %), L1 (7.5 mol %)	25
4	NiBr ₂ ·glyme (1 mol %), L1 (1.5 mol %)	<5

Question 9: Since thermal conditions using Mn as the reductant has shown the advantage in some cases, it seems unfair to claim it in line 105. Please revise it properly. In addition, please specify the source for Mn, and nickel source.

Our response: Thanks for this great suggestion. We revised the sentence: “Interestingly, this electrochemical reductive coupling demonstrates complementary reactivity with traditional reductive coupling methods. When manganese (Mn) as a reductant in the traditional approach, it shows promise for reactions involving pyridine iodides but struggles with the efficient conversion of pyridine bromides and other bromobenzene (Figure 1A and 1B).”

Additionally, we have included the reagent information to Supplementary Notes in SI. Nickel (II) bromide ethylene glycol dimethyl ether ($\geq 97\%$) was purchased from Bidepharm. Manganese (99.8%) was purchased from Adamas-beta and was used as received.

Question 10: As for supplementary table S8. screening of additive, please show the detail. As for the SI, please specify how to reuse the electrodes.

Our response: We have added details to supplementary table S7 and S8. In addition, we have added the “Cleaning and Reuse of Pt electrode” in the SI. “After the reaction is finished, put the Pt electrode into a solution of concentrated hydrochloric acid: ethanol = 1:100 and ultrasonicate for 10 minutes, then wash it with ethanol and acetone and dry it, then reuse it in the next reaction”.

Supplementary Table S7. Increase substrate steric hindrance---methyl to ethyl^a

^aReactions were carried out with **1a** (2.0 equiv.), **2a** (0.2 mmol), NiBr₂•glyme (10 mol %), **L1** (15 mol %), Et₃N (4.5 equiv.), TBABF₄ (1 equiv.), DMAc:THF = 1:1 (2 mL), Pt (1.0 x 1.0 cm²) as the anode. Ni form (1.0 x 2.5 cm²) as the cathode, 4 mA, Room temperature, 6 h. ^bYields were determined by ¹H NMR using CH₂Br₂ as an internal standard. ^cThe ee values were determined by HPLC on a chiral stationary phase.

Supplementary Table S8. Add 4 Å molecular sieves^a

^aReactions were carried out with **1a** (2.0 equiv.), **2a** (0.2 mmol), NiBr₂•glyme (10 mol %), **L1** (15

mol %), Et₃N (4.5 equiv.), TBABF₄ (1 equiv.), 4Å MS (50 mg), DMAc:THF = 1:1 (2 mL), Pt (1.0 x 1.0 cm²) as the anode. Ni foam (1.0 x 2.5 cm²) as the cathode, 4 mA, Room temperature, 6 h.
^bYields were determined by ¹H NMR using CH₂Br₂ as an internal standard. ^cThe ee values were determined by HPLC on a chiral stationary phase.

Reviewer: 2

Question 1: The authors cited e.g., Zhan Lu's enantioselective benzylic arylation via photoredox and nickel dual catalysis, this work utilized an almost similar ligand to the one reported in Lu's work. What is the difference between the two works?

Our response: Thanks for this comment. The biggest difference between the two works is that electrochemistry has a wider substrate range. We applied the reaction conditions of the previous method to this reaction and found that when the substrate was chlorobenzene, the yield was only <5%, and there was a large amount of chlorobenzene left. When the substrate was aryl triflate, no product was generated as illustrated in the following table.

Entry	Reducing reagents	NMR yield (%)	ee (%)
X = Cl/OTf	Electrochemistry (Standard Conditions)	75/67	87/82
X = Cl/OTf	Photochemistry (Ref. Lu's report)	<5/0	-

Question 2: In table 2. substrate scope, are other electron-poor (hetero)aryl chlorides such as chloro-pyrimidines, pyridazines, pyrazines, or triazines compatible with the optimal reaction conditions?

Our response: Thanks for your comments. We have included the results with the above substrates under standard conditions and found that they could all obtain coupling

products with low yields and ee values, which the highest ee not exceeding 22% as illustrated in the following scheme.

Aryl Halides

Question 3: Plenty of work underwent in terms of ligand design for enantioselective couplings since Reisman's report in 2017 and Lu's in 2019. How does Reisman's procedure from 2017, which relies on a BiOX ligand that was notable for its time, work with the new BiIM system?

Our response: Thank you for your valuable comment. Biox ligands can be used in many types of reactions, but we found that under electrochemical conditions, Biox ligands are inferior to BiIM ligands in terms of reaction conversion rate and chiral control, which may be due to the stability of the metal complex. In addition, due to their electron-rich properties, BiIM ligands lead to stronger metal complex activity, thus enabling the activation of chlorobenzene, which is not available in Biox ligands.

Question 4: Does the use of platinum and the subsequent use of TEA have a negative effect on the catalysis, through catalyst deactivation via coordination of TEA to nickel?

Our response: When TEA was used as the anode sacrificial agent, the enantioselectivity of the reaction was not affected, indicating that TEA did not affect the catalyst. In addition, Platinum can be reused, which reduces the impact of frequent replacement of electrodes under sacrificial electrode conditions, which is of great

significance for large-scale production.